# Prospective Evaluation of Two Wall Orbital Fractures Involving the Medial Orbital Wall: PSI Reconstruction versus PDS Repair—Worth the Effort?

**DOI:** 10.3390/jpm12091389

**Published:** 2022-08-27

**Authors:** Max Wilkat, Karsten Hufendiek, Merve Karahisarlioglu, Maria Borrelli, Christoph Sproll, Majeed Rana

**Affiliations:** 1Department for Oral & Maxillofacial Surgery, University Hospital Düsseldorf, Moorenstr. 5, 40225 Düsseldorf, Germany; 2University Eye Hospital, Hannover Medical School, 30625 Hannover, Germany; 3Department for Ophthalmology, University Hospital Düsseldorf, 40225 Düsseldorf, Germany

**Keywords:** orbital wall fracture, medial orbital wall, orbital reconstruction, patient specific implant, polydioxanone foil

## Abstract

Proper treatment of the two-wall fractured orbit is still controversial. Specifically, there is no consensus on the issue of the necessity of medial orbital wall repair. With anatomically critical structures at risk during the surgical approach, surgeons’ view on the necessity of medial orbital wall repair often is restricted and an aesthetically disturbing enophthalmos is more likely to be accepted. Therefore, treatment options range from leaving the medial wall without repair to reconstruction with autogenous tissue or alloplastic materials, which can lead to moderate to severe side effects. However, emerging technologies such as patient-specific implants (PSI) offer a reliable and anatomically correct reconstruction of the bony orbit. This study aimed to evaluate the outcome of full orbital reconstruction using PSIs compared to only orbital floor repair using PDS (bioresorbable polydioxanone) foils leaving the medial orbital wall untouched in traumatic two-wall orbital fractures. Of all patients treated at the University Hospital of Düsseldorf between 2017 and 2019 who suffered from traumatic orbital fracture, only patients with a two-wall orbital fracture involving both the orbital floor and the medial wall (n = 68) were included. Patients were treated either with a PSI (n = 35) or a PDS foil (n = 33). Primary outcome parameters were ophthalmological disturbances analyzed via clinical investigation and intra-orbital angles, volumes and implant position analyzed with radiological 3D-datasets. While a two-wall reconstruction using PSIs led to a significant improvement of the enophthalmos, the rate of postoperative enophthalmos was significantly increased in cases of only orbital floor repair with PDS foils. Radiologically, a significant reconstruction of the three-dimensional bony orbit succeeded with the simple use of PSIs leading to a significant reduction in the traumatically enlarged orbital volume. PSI also led to a significant reduction in the traumatically enlarged medial angle of the orbit. This was not the case for single-floor repair with PDS foil. The results of this study suggest that complex orbital fractures can be reconstructed at an even higher degree of accuracy with selective laser-melted PSIs than PDS foils. In order to achieve a true to original reconstruction of the bony orbit, surgical treatment of the medial orbital wall can be advocated for in the long term depending on the indication.

## 1. Introduction

In CMF Trauma, the most controversial debate considers the indication for the medial orbital wall repair [1]. The indication for fracture care is based on a combination of radiological and clinical findings [2]. In many cases, the radiologically visible size and localization of the fracture help in decision making for surgical care [3]. There is controversy about the treatment of radiologically small defects, as the reaction of orbital soft tissues is difficult to assess [4,5,6,7]. Kunz et al. proposed a conservative treatment for fractures with a size of smaller than 3 cm^2^ in an otherwise inconspicuous clinic [2]. These “small” defects must be taken into account, as some trapdoor fractures can remain undetected and thus escape radiological diagnostics [8].

Clinical symptoms such as diplopia, motility disorders caused by muscle incarceration and an enophthalmos greater than 2 mm are further criteria for the indication to surgical treatment [9]. The latter is hard to evaluate in the early post-traumatic phase due to swelling and hematomas. Ophthalmological symptoms can occur after some time due to contusion and consecutive intramuscular swelling [2]. Therefore, in cases where emergency intervention does not have to be carried out, a wait-and-see procedure is recommended. However, after a period of 2 weeks the phase for early reconstruction has passed and the constantly advancing, post-traumatic fibrosis of the orbital soft tissue must be taken into account [10].

There is still no consensus in global guidelines regarding a strong indication for medial orbital wall repair. One of the reasons is the complex 3D anatomy of the orbit and reduced intraoperative visualization due to limitations in surgical approaches. Common approaches to the medial orbital wall include the coronal and the trans-/retrocaruncular [11,12] approach. However, these approaches harbor side effects such as scar formation with alopecia and structural damage to the extra-ocular muscles. Another reason lies in the diversity of the material used for orbital damage repair. A wide array of material has been used including autogenous materials such as calvarian split grafts, allogenous material such as lyophilized dura mater or homologous donor bone and alloplastic material such as bioresorbable polydioxanone foils (PDS) and titanium meshes [13]. While intra-operatively bended and preformed titanium meshes are low in cost and easy to use, patient-specific titanium implants offer the highest contour accuracy [14]. With newer emerging technologies [15] it was possible for the first time to reliably reconstruct a two-wall orbital fracture with an almost true to original result considering the unaffected healthy side. However, this highly accurate technique [16] comes with disadvantages such as delayed surgical treatment due to the necessary production time of the PSI and a higher cost, although it has to be recognized that production time as well as costs decrease from year to year [17].

Therefore, this study investigated patients with traumatic two-wall orbital fractures by evaluating the benefit of reconstructing the medial orbital wall with a PSI versus only repairing the orbital floor with a PDS foil.

## 2. Materials and Methods

### 2.1. Study Protocol

In this prospective study, patients with unilateral orbital two-wall fracture involving both the medial orbital wall and the orbital floor diagnosed by 3D imaging (CT or CBCT) who received treatment in the Departments of Oral and Maxillofacial Surgery in the University Hospital of Düsseldorf from 2017 to 2019 were included. In cases with an indication for orbital repair/reconstruction due to diplopia, motility disorders, sensitivity disorders and/or an enophthalmos greater than 2 mm, surgery was performed by reconstructing the fractured orbit either with a two-wall patient-specific implant (n = 35) or by performing only orbital floor repair with a bioresorbable PDS foil (n = 33). Assignment to the respective group was made chronologically. The patients to be cared for surgically in the first half of the study period were provided with PDS foil; the patients in the second half were provided with PSIs. Only patients with a full dataset of pre- and post-operative radiological scans and pre-/postoperative (1 to 3 days before and after surgery, respectively) and follow-up (6 months post-treatment) ophthalmologic investigation were included (see Figure 1).

### 2.2. Work Flow for Orbital Reconstruction with A Patient-Specific Implant

The radiological data (CT scan, minimum 1 mm axial layer thickness) were obtained and after digitally segmenting and mirroring the healthy orbit to the fractures site (iPlan CMF 3.0.5, Brainlab, Munich, Germany), a patient-specific implant was designed (see Figure 2). The decision for the medial wall reconstruction was strongly made in the coronal view. The border was pointed with land marks. Additionally, the posterior bulge was analyzed in the axial view and virtually reconstructed. After transmission of the data to the company KLS-Martin (Tuttlingen, Germany) or Synthes (Umkirch, Germany), production took place in a selective laser melting process using Ti-Alloy Ti6Al4V Grade IV. The PSIs had a thickness of 0.3 mm with a 0.5 mm thick cord on the circumference. The whole work flow from data obtainment to delivery of the ready-to-use product took between 8 to 10 working days.

For referencing purposes, a preoperative CBCT scan with a dental reference splint was obtained which was merged with the planning scan.

During surgery (see Figure 3), temporary fixation of a dynamic referencing star (Brainlab Skull Reference Array) in the temporo-parietal region of the contralateral site was performed. The dental reference splint was inserted and the referencing procedure was performed according to the Brainlab protocol using the four coordinates defined by the dental reference splint.

Depending on the surgeon, the retroseptal-transconjunctival, mediopalpebral or infraorbital approach was adopted. After preparation of the orbital soft tissues, the implant could be inserted. Via trajectory-based navigation the correct positioning was evaluated intra-operatively. After fixation of the PSI with one to two osteosynthesis self-tapping screws (1.2 mm diameter × 5 mm length) at the infraorbital rim, passive motility of the eye was checked by forced duction tests. Surgery was finished by closing the incision of approach. The implant position was evaluated by performing a postoperative CBCT scan.

### 2.3. Work Flow for Orbital Reconstruction with a PDS Foil

Depending on the surgeon, a retroseptal-transconjunctival, medial palpebral or infraorbital approach was performed. After preparation of the orbital soft tissues, the PDS foil (0.15 × 30 × 40 mm, orbital cut, perforated) was adapted to the anatomy of the orbit and the defect size by cutting accordingly and then inserted onto the bony defect ensuring circular overlapping coverage of the defect to prevent dislocation. Passive motility of the eye was checked by performing a forced duction test. Surgery was completed by closing the approach’s incision and evaluated with a postoperative CBCT scan.

### 2.4. Ophthalmological Examination

Of the 104 patients treated surgically, all of them received a pre- and post-operative ophthalmological investigation one to three days before and after surgery, respectively. Moreover, all 104 patients were invited to an ophthalmological follow-up appointment six months after surgery. A total of 68 patients appeared for ophthalmological follow-up. All specialist examinations were carried out by the same ophthalmologist. First, the eyelids were inspected for scars, wound healing disorders and lower eyelid affections such as ectropion or entropion.

The subsequent examination of best corrected visual acuity (BCVA) was carried out using the autorefractometer KR- 800S from Topcon (Topcon Corporation, Tokyo, Japan). For this purpose, the patient should read the displayed number and letter series. The smallest series of numbers that could be easily recognized was noted. Visual acuity was measured in percentages.

The stereo vision test according to Lang and the Bagolini striated glass test were performed to test stereo vision and binocular vision. Ocular motility was assessed in the nine cardinal gaze positions using a fixation light and an Occluder (Oculus, Wetzlar, Germany). Documentation included movement restrictions and diplopia within a field of 30°.

The vertical bulb position was evaluated using Kestenbaum glasses. Based on the millimeter scaling, deviations compared to the contralateral side could be evaluated. If there was a hypo-/hyperglobe, the field for the corresponding page was ticked and the estimated difference in millimeters was specified. The sagittal bulb position was measured using a Naugle exophthalmometer (Oculus, Wetzlar, Germany). By providing support on the superior and inferior orbital frame, the values for the left and right eye as well as the inter-pupillary distance were measured in millimeters. An aesthetically disturbing enophthalmos was defined as a sagittal retral deviation of the eye position of more than 2 mm.

In addition, a blunt sensitivity test of N. infraorbitalis for anesthesia, paresthesia, hypoesthesia and hyperesthesia was carried out by palpation of the skin with a dental probe and a cotton carrier in a side comparison.

### 2.5. 3D-Data Evaluation

All pre- and postoperative CT and CBCT datasets were stored in DICOM format to a local computer in a hard and soft tissue window of the axial plane and a layer thickness between 0.75 mm and 1.25 mm. Subsequently, the raw data were imported to iPlan CMF 3.0.5 from Brainlab^®^ (Feldkirchen, Germany). The program calculated the coronary and sagittal layer for each patient from the axial plane. The soft tissue window was exported to the bone window via the “predefined windowing” option. All datasets were aligned according to the Frankfurt Horizontal and Median Sagittal Plane. With the help of the three-dimensional and the multi-planar view, the symmetrical alignment of both orbits was performed.

The measurement of the defect size was performed with the maximum diameter in the coronary and sagittal plane for the orbital floor and in the coronary and axial plane for the medial orbital wall. The respective distances were expressed in millimeters.

For the healthy and the fractured orbit, respectively, an angle measurement was carried out between the medial orbital wall and the orbital floor in the coronary plane with the following three defined points: the cranial boundary of the medial wall, the transition zone and the transition of the orbital floor to the lateral wall (see Figure 4a). The measurement was carried out for both sides in the anterior, middle and posterior thirds of the orbit. These were defined in the sagittal plane by the anterior bony orbital margin, the posterior bony boundary (the so-called “posterior ledge”) and the middle of this route.

For volume measurement of the healthy and fractured side, both orbital cavities were segmented by the program after selection via the menu item “object creation” (see Figure 4b). The outlines of the enlarged volume of the fractured side were dragged manually via the “smart shaper” option to define the volumes individually.

In addition, the implant position was evaluated in the PSI group. An insufficient position was recognized if the bony defect in the sagittal and coronary layer was not sufficiently covered by the implant, if there was a distance of more than 3 mm from the orbital walls and if the implant did not rest sufficiently on the posterior ledge.

All measurements were carried out three times by two independent investigators to rule out intra- and inter-observer variability.

### 2.6. Statistical Analysis

All data collection and storage was carried out using Excel spreadsheets (Excel 14.0, Microsoft Corporation, Washington, DC, USA). The statistical evaluations were carried out with the SPSS^®^ software (SPSS version 25.0, IBM SPSS, Chicago, IL, USA). In order to record the statistical relationships between the groups and the times, a multifactorial analysis of variance was carried out with repetition of measurements on one factor. The relationship of orbital parameters with regard to the mode of therapy was carried out by performing binary logistic regression, chi-square tests, *t*-tests and McNemar tests. Results with a *p*-value of less than 0.05 were considered significant.

## 3. Results

### 3.1. Demographic

Of a total of 68 enrolled patients, 73.5% were male (n = 50) and 26.5% female (n = 18). The average age at the time of trauma was 45.6 years (SD = 20.54) for the PSI group and 43.0 years (SD = 19.95) for the PDS group, respectively. The right orbit was affected in 33 cases while the left side was affected in 35 cases. The most common cause of trauma was a crudity offence which accounted for 44.0% (n = 33), followed by falls with 26.5% (n = 18). Trauma causes were equally distributed across the study groups (PSI vs. PDS) without significant difference (see Figure 5).

The average operation time from cut to suture was 72.11 min (SD = 28.99) in group PSI and 67.39 min in group PDS (SD = 37.64) without a significant difference. The time passed from trauma to surgical treatment was significantly longer for the PSI group with 20.03 days (SD = 15.77) compared to 6.52 days (SD = 3.50) for the PDS group (*p* = 0.00163).

### 3.2. Clinical Parameters

Pre-operative diplopia was found in a total of 27 patients (PSI = 13, PDS = 14—see Figure 6a). In the long-term follow-up 6 months after surgery this had significantly improved, with three cases of lasting diplopia for each study group. Strikingly, the post-operative occurrence of diplopia directly after the surgical procedure rose significantly in cases of PSI implantation (n = 19), while for PDS cases diplopia had already been reduced post-operatively in 11 cases (Pearson Chi-Square: df = 2.00, *p* = 0.016).

Globe malposition was observed in fifteen patients in the PSI group pre-operatively and could be corrected in fourteen cases leaving only one patient with hyperglobe after PSI implantation (see Figure 6b). In the PDS group, a pre-operative globe malposition was found in seven cases, while after surgery in the long-term follow up a total of twenty three patients suffered from aesthetically impairing globe malposition. A two-wall reconstruction via PSI implantation could significantly reduce malposition of the globe (*p* = 0.0003), while orbital floor repair with PDS foil significantly worsened the globe malposition from the pre-operation stage to follow-up (*p* = 0.0019). This was also the case for enophthalmos correction evaluated on its own (*p* = 0.0063 for PSI, *p* = 0.0003 for PDS).

While five patients suffered from pre-operative hypoesthesia in the PSI group, the cases of pre-operative hypoesthesia was two times higher in the PDS group with a total of ten patients (see Figure 6c). In both groups, the rate of hypoesthesia increased post-operatively. In the long-term follow-up, the rate of sensibility disorders increased by 20% in both groups compared to the pre-operative status, while in the PSI group one patient suffered from hyperesthesia and one patient from paresthesia.

### 3.3. Radiological Parameters

Concerning intra-/inter-observer variability, the Kruskal–Wallis H test revealed no statistically significant difference between the intra-observer and inter-observer groups for both the PSI and the PDS group.

Defect sizes between the groups were comparable with no significant difference except for the sagittal floor diameter which was 15.26 mm (SD = 6.87 mm) for the PSI group and 11.21 mm (SD = 6.20 mm) for the PDS group (see Figure 7a). The calculated total defect area was 2.48 cm^2^ for the PSI group and 1.93 cm^2^ for the PDS group.

The overall mean anterior angle of the orbit did not differ considerably between the unaffected healthy side (=120.42°), which served as a template, and the fractured site pre- (120.37°) and postoperatively (120.50°). The same was true for the posterior angle, which was 134.40° for the unaffected side, 131.38° for the fractured site pre-operatively and 131.65° post-operatively. However, the mean medial angle, which was 123.19°, differed in particular for the PSI group. It was significantly reduced on the fractured site from 117.06° pre-operatively to 124.83° post-operatively (*p* = 0.001) (see Figure 7b). This was not the case for the PDS group, which showed a medial angle pre-operatively of 113.84° and insignificant change for the medial angle post-operatively which was 112.54° (*p* = 0.656).

The overall average volume of the healthy orbit was 30.50 mL (SD = 3.04 mL), while the volume of the fractured side at diagnosis was enlarged at 33.18 mL (SD = 3.53 mL). After receiving therapy, the average volume was 31.85 mL (SD = 3.34 mL). The mean orbital volume for the PSI group viewed separately was 33.56 mL (SD = 3.64 mL) pre-operatively and could be significantly reduced by surgery to 31.60 mL (SD = 3.17 mL) (*p* = 0.019) (see Figure 7c). The mean orbital volume for the PDS group was 32.76 mL (SD = 3.47 mL) pre-operatively and was not significantly different post-operatively with a volume of 32.11 mL (SD = 3.60 mL) (*p* = 0.459).

Concerning PSI malpositioning, four cases fulfilled the defined criteria. Of these, only one patient (2.86%) suffered from globe malposition in the form of a moderate hyperglobe of 2 mm. None of the four patients suffered from diplopia, other forms of globe malposition or sensibility disorders.

## 4. Discussion

The most common symptoms of orbital fractures include diplopia, enophthalmos and sensibility disorders [18,19,20]. Studies to predict these clinical appearances refer, among other things, to the radiological defect size in order to be able to make a decision regarding the surgical indication [2]. Ploder et al. showed in their study that there is a significant relationship between radiological fracture expansion and enophthalmos as well as diplopia [21,22].

The groups of surgically cared for patients with PSI (n = 35) and PDS foil (n = 33) in this work offered approximately comparable moderate defect sizes (2.48 cm^2^ for PSI group and 1.93 cm^2^ for PDS group). While the pre-operative rate of diplopia was comparable as well between the groups (13 for PSI group, 14 for PDS group), pre-operative globe malposition was present over 2 times more often in the PSI group (15 for PSI group, 7 for PDS group).

A comparison of PSIs and PDS foils is not described in the literature. The isolated consideration of bioresorbable materials shows that the volume differences are not significant pre- and post-operatively [23,24]. You et al. found healed orbital bones in radiological aftercare, which were asymmetrical and had a volume increase compared to the healthy opposite side [24]. In the current study, radiological aftercare was not routinely performed; however, in the postoperative radiological datasets the same insufficient volume reduction for the PDS group could be observed while the PSI group showed a significant volume reduction.

Kontio et al. examined 16 patients treated with PDS foil and observed an insufficient form and volume reduction despite radiologically visible bone growth [23]. Preoperatively, two people had an enophthalmos which increased to six patients with enophthalmos in the aftercare routine. Initial double images and sensitivity disorders improved, which is in accordance with the current study. They concluded that due to flexibility and degradation after just 2 months, PDS foils cannot be successfully used for reconstruction and bone healing is affected [23,25]. These studies do not relate the defect size to the observed symptoms. However, this relation was observed by Baumann et al., as an influencing factor [26]. Out of thirty two people, seven had an enophthalmos post-operatively, of which five out of six patients had a major defect with a size of larger than 2.5 cm^2^. Baumann et al. concluded that PDS foils do not have sufficient stability to maintain the orbital content in case of large defects. However, they stated that “blow-out and midfacial fractures with small to moderate defects in the orbital floor (up to a size of 2.5 cm^2^) can be reconstructed by PDS foil to avoid enophthalmos” [26].

Our patient collective offered a two-wall defect with a mean defect size of 2.21 cm^2^. In contrast to the study of Baumann et al., we found a significant increase in post-operative enophthalmos for the PDS group despite a smaller defect size than described by Baumann et al. However, in our study the medial wall was affected by the fracture. This leads us to the conclusion that whenever there is a combined defect of the medial wall and the floor, sole floor repair with PDS foil is insufficient even if the combined defect size is smaller than 2.5 cm^2^. On the contrary, a defect reconstruction with a two-wall PSI led to a significant reduction in the orbital volume followed by a significantly reduced rate of enophthalmos in the follow-up.

However, the simultaneous use of PSIs and intra-operative navigation might be considered as a limitation of the study as in the PDS group no navigation was applied. There are many studies underlining the precision of orbital reconstruction by using intra-operative navigation [13,27,28]. Thus, one might discuss that the observed increased precision in orbital reconstruction for the PSI group is not achieved by the PSI itself but by the navigation. However, in our study only implant positioning and not the orbital outlines was considered during intra-operative navigation. It is well-known that with the help of navigation-assisted surgery and patient-specific implants, the most precise reconstruction of the orbit is currently possible [15,29,30,31,32]. Gander et al., showed in their study that no patient with PSI (n = 12) needed a second intervention. Furthermore, no visual impairments could be detected [33]. In studies conducted by Schönegg et al., with 41 PSIs inserted, a satisfactory position was found for each implant on the postoperative radiological scan [34]. Zimmerer et al. examined the X-ray implant shape and location with regard to clinical parameters in a multi-center study. No significant link could be established between undesirable consequences and implant position [35]. These results coincide with our study analyzing two-wall orbital fractures. Here, one of the thirty five included patients with PSI implantation had an undesired implant position. Undesirable results can correlate with the experience of the treating surgeon with navigation-assisted surgery or to the intra-operative difficulty of implant positioning. No significant relationship between a different implant position and clinical parameters could be established. In these results, the influence of orbital soft tissue reaction was not considered. Despite accurate reconstruction of the orbit, temporary postoperative consequences are unavoidable [35]. The overall small number of mispositioned implants tends to show the reliability of computer-assisted procedures [15,16,36,37].

## 5. Conclusions

According to the research, medial wall reconstruction in two-wall orbital fractures is important in order to avoid long-term consequences such as an aesthetically disturbing enophthalmos. The difficulty of reconstructing this area lies in the demanding, three-dimensional anatomy of the bony orbit. In addition, there are critical structures such as the ethmoidal vessels and the optic nerve, which present a challenge for the surgeon due to poor visibility in the operating field. Far-reaching consequences such as motility disorders, muscle entrapment, bleeding and dislocation of the implant require a safe method for patients and practitioners.

Patient-specific implants in combination with intra-operative navigation and a transconjunctival-retroseptal approach tend to represent reliable therapy. In addition to a significant volume reduction, PSIs are a safe method to achieve a significant restoration of the three-dimensional anatomy. Despite higher costs and a longer trauma-to-surgery period due to the necessity of the production time, one can advocate for the use of PSIs over PDSs even in small combined defects of two-wall orbital fractures as a personalized, patient-centered solution.

## Figures and Tables

**Figure 1 jpm-12-01389-f001:**
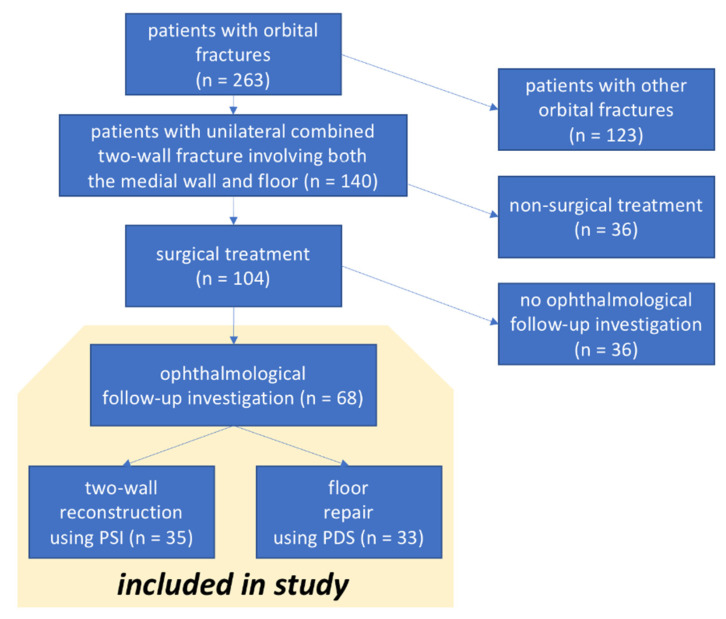
Composition of study group.

**Figure 2 jpm-12-01389-f002:**
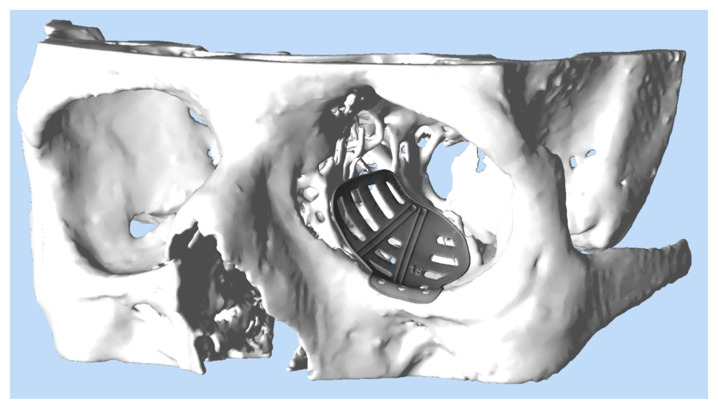
Virtual design of a PSI for reconstruction of the left orbit. The PSI (grey) has been provided with trajectory lines for intraoperative navigation and drain grooves to ensure postoperative drainage.

**Figure 3 jpm-12-01389-f003:**
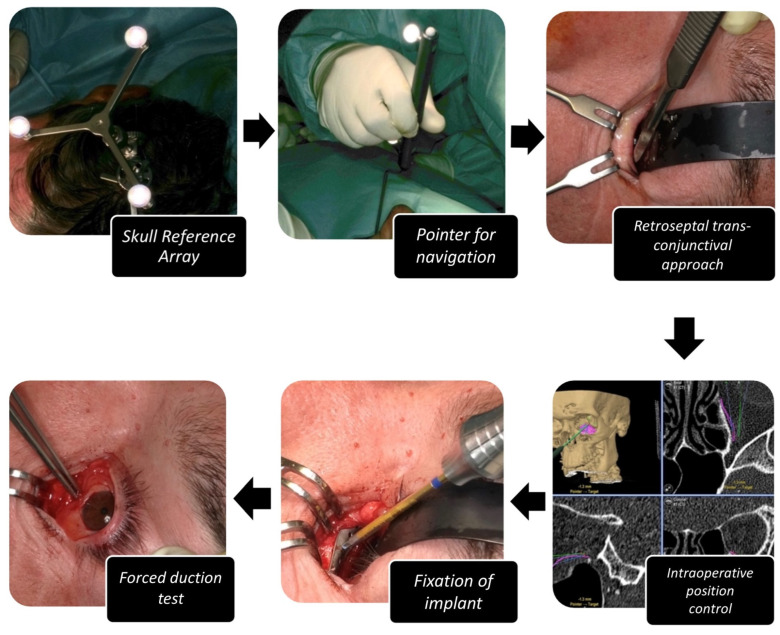
Operation procedure for computer-assisted, navigated PSI implantation for orbital reconstruction. The individual steps of set-up of navigation, approach, placement, fixation and placement control of PSI including forced duction test are shown.

**Figure 4 jpm-12-01389-f004:**
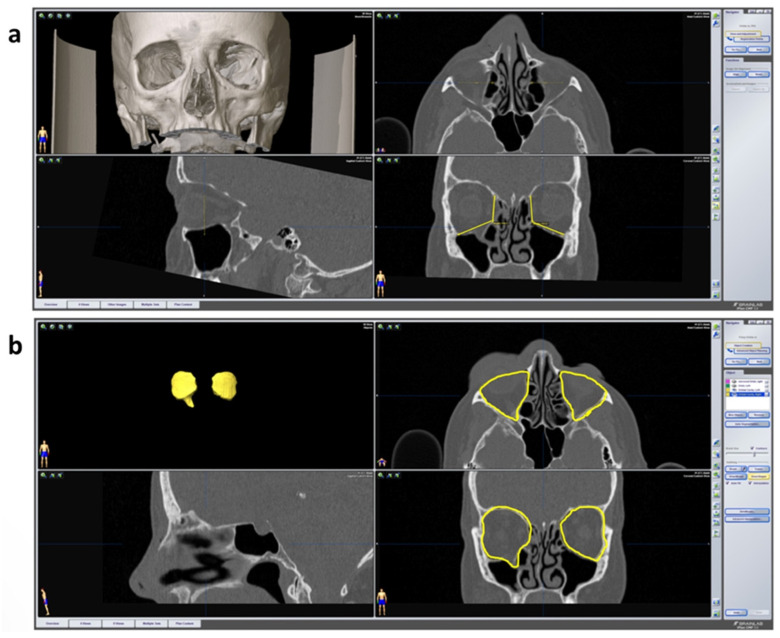
Radiological parameters. (**a**) Angle measurement between the medial orbital wall in iPlan CMF 3.0.5. in coronary layer. The datasets are shown in multi-planar view including 3D reconstruction, axial, sagittal and coronary plane. At the bottom right, the angles of both orbits can be seen in yellow. The values are automatically calculated by the program and displayed. (**b**) Orbital cavity algorithm for automatic volume calculation in iPlan CMF 3.0.5. At the top left the segmented orbital cavities are displayed in 3D volume. On the upper right (axial) and bottom right (coronary), the outlines of the orbits are shown in yellow created via the “smart shaper” function. The right orbital content herniates into the maxillary cavity (“drop-shape”) due to the fracture of the orbital floor.

**Figure 5 jpm-12-01389-f005:**
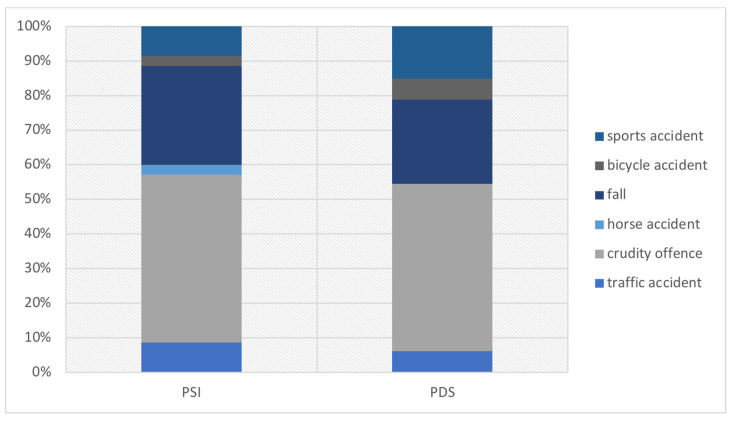
Trauma cause for orbital fractures differentiated by surgical treatment.

**Figure 6 jpm-12-01389-f006:**
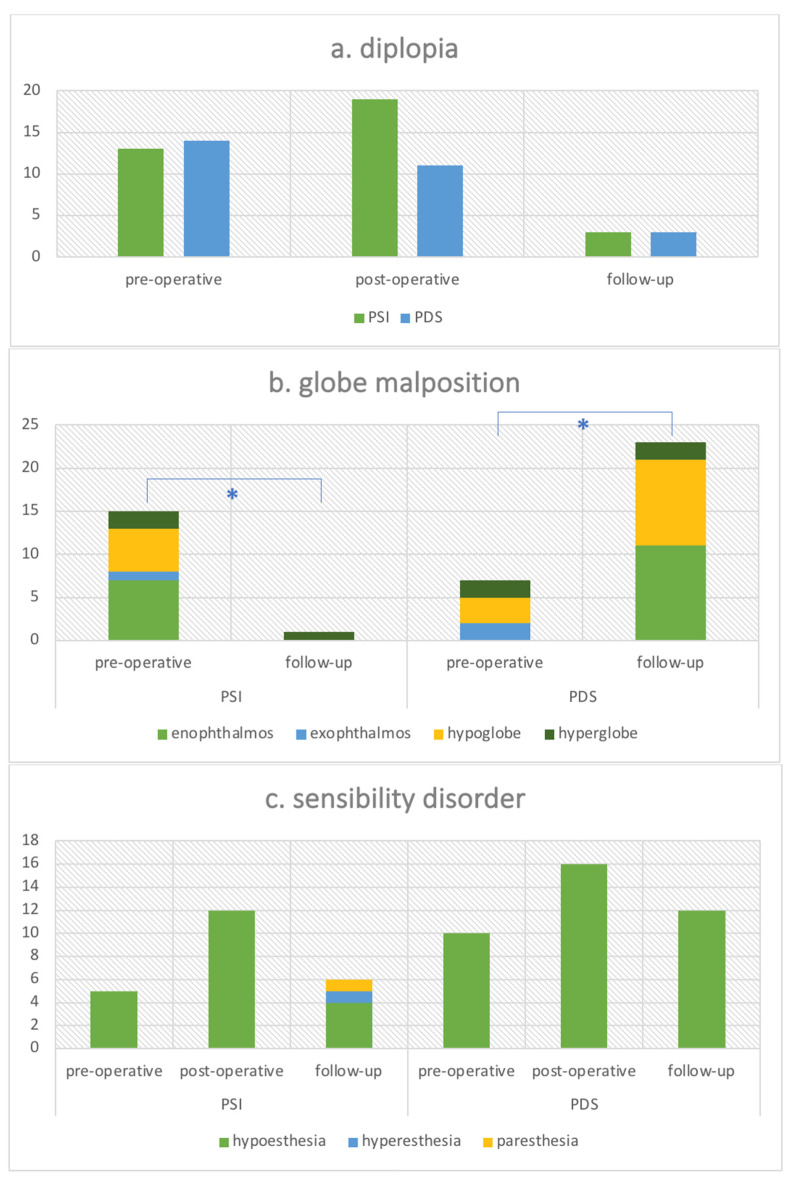
Clinical signs in two-wall orbital fracture. (**a**) Number of cases of diplopia pre-operative, post-operative and in the follow-up for the different study groups “PSI” versus “PDS”. (**b**) Number of cases of malposition of the globe including enophthalmos, exophthalmos, hypoglobe and hyperglobe pre-operative and in the follow-up for the different study groups “PSI” versus “PDS”. * indicates significance with *p* < 0.05. (**c**) Number of cases of sensibility disorders including hypoesthesia, hyperesthesia and paresthesia pre-operative, post-operative and in the follow-up for the different study groups “PSI” versus “PDS”.

**Figure 7 jpm-12-01389-f007:**
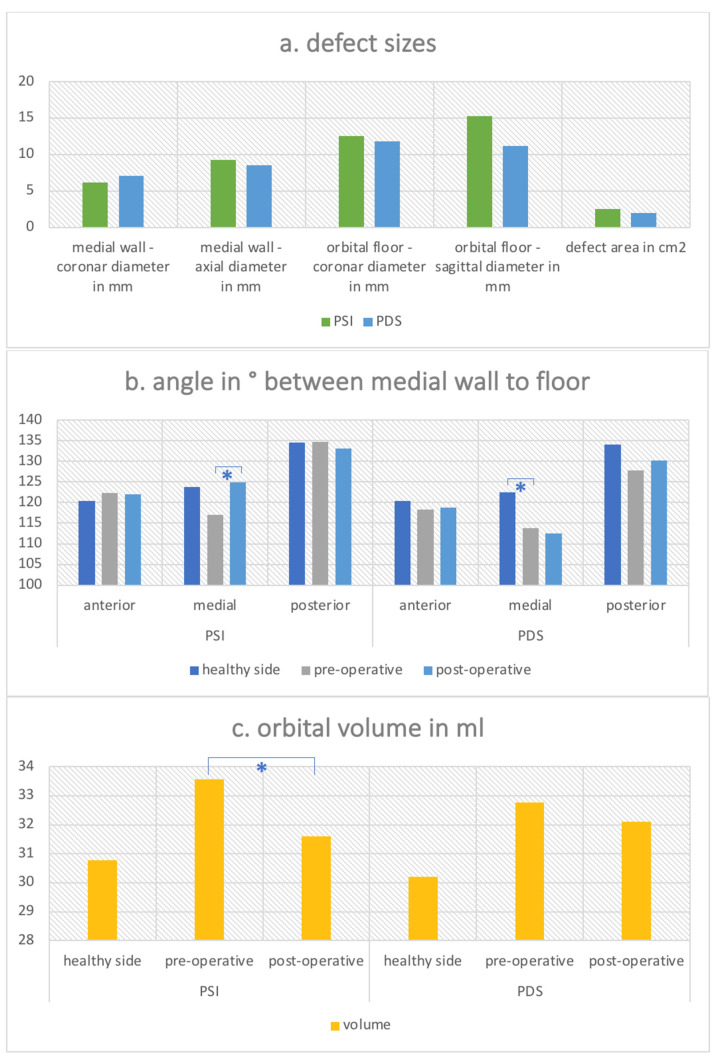
Radiological parameters in two-wall orbital fracture. (**a**) Defect sizes for the different study groups “PSI” versus “PDS”. Two diameters in mm are displayed for each medial wall defect and the floor defect as well as the overall defect area in cm^2^. (**b**) Angles in ° are displayed for each study group “PSI” versus “PDS” measured in the anterior, medial and posterior part of the orbit for the healthy unaffected side as well as the fractured side pre- and post-operatively. * indicates significance with *p* < 0.05. (**c**) Orbital volume in ml for the different study groups “PSI” versus “PDS”measured for the healthy unaffected side as well as the fractured side pre- and post-operatively. * indicates significance with *p* < 0.05.

## Data Availability

Not applicable.

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
