# Peer review of "Prospective Evaluation of Two Wall Orbital Fractures Involving the Medial Orbital Wall: PSI Reconstruction versus PDS Repair—Worth the Effort?"

_jpm, 2022, doi:10.3390/jpm12091389_

Round 1

Reviewer 1 Report

A well conducted and thoroughly written study contemporary to the recent advances in management of the orbital fractures but still needs several corrections.

This study is dedicated to the indications and methods of medial orbital wall reconstruction, which is located at the gray area of the modern knowledge. However, several landmarks of it, like importance of navigation, is well described in a literature. I propose acceptance of it with minor revisions.

 TITLE

Fine.

Abstract.

Deeper clarification of the conclusions could enrich the article. Inclusion of the quality of life evaluation statements disguises the main goal of the study (see comments to “Result” section).

Introduction.

No doubt, that the problem of two wall, including medial one, orbital reconstruction is very actual, as well as controversial and comparison of the PSI and PDS is absolutely correct. However, it should be recognized that the production time of the PSI and a higher cost leads to be decreased from year to year. The time period from ordering the PSI to its delivery to the hospital could be less than 3 days (Chepurnyi Y, Zhukovtseva O, Kopchak A, Kanura O. Clinical application of automated virtual orbital reconstruction for orbital fracture management with patient-specific implants: A prospective comparative study [published online ahead of print, 2022 Jun 3]. J Craniomaxillofac Surg. 2022; S1010-5182(22)00069-5. doi: 10.1016/j.jcms.2022.05.006 ) , which is absolutely enough for emergency reconstruction and it could be considered as a method of choice for routine orbital reconstruction.    

Materials and methods.

The main question of the described CAD Protocol is how the border of PSI at the medial orbital wall was defined? Medial orbital wall is often misrepresented in the CT and is difficult for segmentation. These areas often appear as holes in the 3-D data set, even though the bone is intact. Therefore, these holes are often called pseudoforamina. (Wagner, M.E.H., Gellrich, NC., Friese, KI. et al. Model-based segmentation in orbital volume measurement with cone beam computed tomography and evaluation against current concepts. Int J CARS 11, 1–9 (2016). https://doi.org/10.1007/s11548-015-1228-8). Due to this, as well as to subperiosteal medial wall fragmentation, misrepresented in the CT, true size of the defect could be over- or underestimated.  So the location of the PSI border at the medial orbital wall is very important in a relation to the true –to –original reconstruction.       

 Also, such details as thickness of the PSI, kind of titanium, type of screws possibly could be interesting for the readers.

Inter and intra-observer differences should be evaluated when we describe the estimation of the bony defect size, based on CT segmentation.     

Results.

The part of this section, related to the effects of surgical approaches for orbital management, should be considered for exclusion form the article. The fact, that the external approaches (mediopalpebral + infraorbital) having been used nearly 3 times more often in the PDS group compared to PSI group, could influence the esthetic evaluation of the surgical outcome, but at the same time could not effect the precision of the orbital shape reconstruction and functional results of surgery. From the same point of view and taking to the account importance to maintain an appropriate quality of live after surgery, my suggestion is to exclude evaluation of QoL after trauma from this study and to prepare separate article, dedicated to this topic. The answers of the 11 patients from the 104 possibly could form the basis for bias.

Discussion.

Good written section however, as for my opinion, the application of the intraoperative navigation should be considered as limitation of this study, because there are a lot of studies, suggested the preference of the navigation for achieving precise orbital reconstruction, especially comparing with such simple procedure as PDS foil one-wall surgical reconstruction.      

Conclusions.

Fine.

Reviewer 2 Report

Authors did study to compared patient specific implant vs foils (bioresorbable polydioxanone) in treating concomitant medical orbital and floor fractures. This paper has a merit for publication however, there are several major issues should be addressed before acceptance

1.       Title:  too long title and abbreviation should be removed

2.       Abstract: too long and complicated, authors should rewrite it in a simple and understandable way (only important info should be mentioned)

3.       Introduction:  too long. Authors should mention clarification of abbreviation at the first time.

4.       There was no connection between the sentences and paragraph.

5.       Authors should be focused on the last paragraph of introduction (rational of study)

6.       Its mandatory to mention strong rationale of study (why it’s important to do this study)

7.       Method: authors should remove patients with others orbital fractures and those patients treated by non-surgical ttt from figure 1

8.       Its mandatory to mention clear inclusion criteria

9.       Authors never mentioned type pf study if it was retrospective or prospective  

10.   Authors should mention details about sample size calculation (otherwise put it as limitation of study in discussion)

11.   The whole paper should be shortened

12.   Extensive editing of English language and style required
